# Oxidative Stress Modulation by ncRNAs and Their Emerging Role as Therapeutic Targets in Atherosclerosis and Non-Alcoholic Fatty Liver Disease

**DOI:** 10.3390/antiox12020262

**Published:** 2023-01-24

**Authors:** Jorge Infante-Menéndez, Paula González-López, Raquel Huertas-Lárez, Almudena Gómez-Hernández, Óscar Escribano

**Affiliations:** Hepatic and Vascular Diseases Laboratory, Biochemistry and Molecular Biology Department, School of Pharmacy, Complutense University of Madrid, 28040 Madrid, Spain

**Keywords:** oxidative stress, atherosclerosis, NAFLD, ncRNAs, miRNAs, lncRNAs, circRNAs, RNA-based therapies

## Abstract

Atherosclerosis and non-alcoholic fatty liver disease (NAFLD) are pathologies related to ectopic fat accumulation, both of which are continuously increasing in prevalence. These threats are prompting researchers to develop effective therapies for their clinical management. One of the common pathophysiological alterations that underlies both diseases is oxidative stress (OxS), which appears as a result of lipid deposition in affected tissues. However, the molecular mechanisms that lead to OxS generation are different in each disease. Non-coding RNAs (ncRNAs) are RNA transcripts that do not encode proteins and function by regulating gene expression. In recent years, the involvement of ncRNAs in OxS modulation has become more recognized. This review summarizes the most recent advances regarding ncRNA-mediated regulation of OxS in atherosclerosis and NAFLD. In both diseases, ncRNAs can exert pro-oxidant or antioxidant functions by regulating gene targets and even other ncRNAs, positioning them as potential therapeutic targets. Interestingly, both diseases have common altered ncRNAs, suggesting that the same molecule can be targeted simultaneously when both diseases coexist. Finally, since some ncRNAs have already been used as therapeutic agents, their roles as potential drugs for the clinical management of atherosclerosis and NAFLD are analyzed.

## 1. Introduction

Oxidative stress (OxS) is a source of cellular stress caused by an imbalance between the production of reactive oxygen species (ROS) and their neutralization by cellular antioxidant machinery. When present at physiological levels, ROS finely tune processes such as transcription, the cell cycle, or apoptosis [1]. However, an excess of ROS leads to OxS, which has been related to pathogenic signaling and several diseases, including cancer, Alzheimer’s disease, obesity, type-2 diabetes mellitus (T2DM), metabolic syndrome (MS), atherosclerosis, and non-alcoholic fatty liver disease (NAFLD) [1,2]. OxS drives disease progression due to the deleterious effects of ROS overproduction, which alter and damage all types of biological molecules. Among these, lipid peroxidation, membrane instability, protein, and mitochondrial DNA (mtDNA) alteration are responsible for further cellular and tissue damage [3].

Multiple organelles generate ROS in the cell, such as lysosomes, the endoplasmic reticulum, and mitochondria. These compartments contain several enzymes that may produce ROS themselves, such as nitric oxide synthase (NOS), NADPH oxidase, xanthine oxidase, myeloperoxidase (MPO), lipoxygenase (LOX), and cyclooxygenase (COX) [4]. Consequently, cells also possess several antioxidant strategies to counteract ROS production, which include enzymes such as superoxide dismutase (SOD), catalase, glutathione peroxidase (GPx), and thioredoxin reductase (TrxR), as well as molecules such as glutathione, coenzyme Q, and various vitamins, such as C, A, or B [5,6].

OxS plays a relevant role in obesity and most of its chronic comorbidities, such as MS, insulin resistance (IR), NAFLD, or atherosclerosis. Obesity is also related to low-grade chronic systemic inflammation due to excessive fat accumulation that affects the adipose tissue and promotes the secretion of pro-inflammatory adipokines [7]. As a matter of fact, adipokines can increase ROS production, promoting OxS and further stimulating the aberrant adipokine secretion [8]. This state of OxS in the context of long-term obesity is exacerbated by the depletion of antioxidant sources [4,7].

Both types of cellular stress, inflammation and OxS, can lead to IR and hyperglycemia through the activation of NF-κB and JNK and play a crucial role in the progression of both atherosclerosis and NAFLD [6,8,9,10]. On one hand, adipocytes can secrete several pro-inflammatory adipokines that promote endothelial dysfunction, leukocyte adhesion, and immune system cell activation (e.g., T-cells, monocytes, neutrophils, and macrophages), including leptin, tumor necrosis factor alpha (TNF-α), or interleukin 6 (IL-6) [7,8]. In a dyslipidemic and pathologic environment, low-density lipoprotein (LDL) can infiltrate and accumulate in the subendothelial space, where it is susceptible to becoming oxidized (ox-LDL) [11]. Ox-LDL endocytosis, in conjunction with endothelial NOS (eNOS) uncoupling and the hyperactivation of NADPH oxidases (NOXs), can increase ROS levels, ultimately contributing to the atherosclerotic process by enhancing smooth muscle cells and macrophage migration, their transformation into foam cells, collagen deposition, and the release of matrix metalloproteinases (MMPs) [11] (Figure 1).

On the other hand, OxS is postulated to be critical in the onset of NAFLD and its progression to non-alcoholic steatohepatitis (NASH), a more severe form of the disease defined by the presence of steatosis with inflammation and fibrosis [12,13]. When the liver suffers an oversupply of free fatty acids (FFAs), this leads to excessive electron flow through the electron transport chain (ETC). This process, known as mitochondrial dysfunction, increases the formation of ROS and generates OxS [9,14]. Additionally, peroxisomes may also undergo an aberrant activation due to the disproportionate levels of FFAs, which in turn increases ROS production [15]. Finally, the pro-inflammatory environment can activate Kupffer cells (KCs) and hepatic stellate cells (HSCs), promoting liver damage and fibrosis, both distinguishing features of NASH [16] (Figure 1).

It is worth stressing that patients diagnosed with NAFLD present a higher risk of developing cardiovascular disease, most notably atherosclerosis [17]. Although the pathophysiological mechanisms that link these two diseases are not fully understood, several signaling pathways and molecular mediators are well-known to drive the joint progression of both diseases [18]. As mentioned before, the activation of NF-κB, the secretion of pro-inflammatory cytokines such as TNF-α or IL-6, and the hyperactivation of NOX enzymes are common alterations in atherosclerosis and NAFLD, which promote OxS and inflammation, consequently favoring the aggravation of both pathologies. In this regard, the need for effective treatments against these diseases becomes evident. During the last few years, the participation of non-coding RNAs (ncRNAs) in the regulation of metabolism, as well as in diseases such as obesity, NAFLD, or atherosclerosis, has been extensively described [3,19]. ncRNAs are functional RNAs transcribed from DNA that do not encode proteins. They are classified into two groups depending on their sizes: long non-coding RNAs (lncRNAs) if they are greater than 200 nucleotides and small non-coding RNAs if they are less than 200 nucleotides [20]. Additionally, small ncRNAs can be subclassified as microRNAs (miRNAs), small interfering RNAs (siRNAs), small nuclear RNAs (snRNAs), small nucleolar RNAs (snoRNAs), piwi-interacting RNAs (piRNAs), transfer RNAs (tRNAs), and ribosomal RNAs (rRNAs) [20]. In addition, circular RNAs (circRNAs) represent a novel type of ncRNA with special characteristics due to their structure, and were also related to the pathogeneses of various diseases, such as diabetes, cardiovascular diseases, or cancer [21].

LncRNAs can exert different regulatory functions, depending on their subcellular localization. Nuclear lncRNAs regulate gene expression by activating or repressing gene transcription and by epigenetic processes, such as chromatin remodeling. Cytoplasmic lncRNAs can also modulate post-transcriptional gene expression through direct interaction with complementary mRNAs, acting as sponges for miRNAs or facilitating protein ubiquitination [19]. Regarding short ncRNAs, the role of miRNAs as regulators of gene expression has been widely described in the last decades. Their main function is to downregulate the expression of their target mRNAs by interfering with the translation machinery or inducing their degradation [3]. Finally, circRNAs also participate in gene expression regulation, modulating alternative splicing or operating as miRNA sponges [21].

LncRNAs, miRNAs, and circRNAs have been related to OxS and ROS production in pathologies such as atherosclerosis and NAFLD [21,22]. The therapeutic applications of ncRNAs go beyond their use as diagnostic and prognostic biomarkers since they can be used as targets or tools in gene therapy [23,24]. This review provides an overview of the most recent advances in ncRNA pathogenic mechanisms and therapeutic strategies regarding OxS in the fields of atherosclerosis and NAFLD.

## 2. Oxidative Stress Modulation by ncRNAs and Its Effect on Atherosclerosis

Atherosclerosis was at first considered an industrialized country disease, but nowadays it has become the main cause of death in the world, which generates the urge to better understand and treat this malady [25]. During the onset and progression of atherosclerosis, hypercholesterolemia and inflammation are pivotal [26], but other processes, such as OxS, also contribute to the severity of the disease [27]. In the three main cell types in vascular tissues, endothelial cells (ECs), vascular smooth muscle cells (VSMCs), and macrophages, OxS plays a role in promoting atherosclerotic progression [27].

### 2.1. Effect of Oxidative Stress and ncRNAs in Endothelial Cells during the Progression of Atherosclerosis

It is commonly known that, in the early stages of the atherosclerotic process, ECs are exposed to ox-LDL, promoting endothelial dysfunction by a mechanism mediated by the nitric oxide synthase pathway [27]. In fact, vascular damage is increased by eNOS uncoupling, higher superoxide production by NOX enzymes, and mitochondrial dysfunction [28], leading to the progression of atherosclerosis (Figure 2). All these pathways have been widely studied; however, the nuclear factor erythroid-2 related factor 2 (Nrf2)/heme oxygenase (HO-1) pathway has emerged as crucial in recent years. Nrf2 acts as a transcription factor in the nucleus and promotes the expression of antioxidant enzymes, such as HO-1, which was proved protective against atherosclerosis in ApoE^-/-^ mice models [29].

Newly, shear stress has surfaced as an important modulator of endothelial dysfunction [30]. Not only were eNOS expression and nitric oxide (NO) production increased in ECs exposed to shear stress [29], which led to ROS production and OxS, but there were also other pathways activated by this stress (Figure 2). Some of these pathways participated in the generation of OxS, such as bone morphogenetic protein 4 (BMP4) or hypoxia-inducible factor HIF-1α [30].

ncRNAs participate in ECs during atherosclerotic progression at different levels. For example, the lncRNA sONE is induced by hypoxia in ECs and negatively modulates eNOS [31,32], disbalancing ROS production and increasing OxS. MALAT1 is another lncRNA that is induced in ECs by OxS, but the effect this lncRNA has is unknown. Some studies attribute to MALAT1 a promoting role in oxLDL-driven inflammation and OxS by regulating the miR-181b/thymocyte selection-associated high-mobility group box (TOX) axis since increases in MALAT1 and TOX by ox-LDL promoted ROS production, NOX activity, and TNFα expression in ECs [33]. Others suggest a protective role of this ncRNA against oxLDL-driven stress by competing with miR-22-3p, as MALAT1 modulated CXCR2, a miR-22-3p target that protects ECs against ox-LDL-induced injury [34] (Figure 2).

However, most of the known ncRNAs that promote atherosclerosis in ECs are microRNAs (miRNAs). miR-34a increased ox-LDL-treated EC apoptosis by targeting histone deacetylase (HDAC) 1, which promoted the expressions of pro-apoptotic proteins, such as pro-caspase-3 and pro-caspase-9 [35]. In fact, ROS can increase or modulate the expressions of miRNAs: for instance, miR-155 inhibition increased autophagy in H_2_O_2_-treated human umbilical vascular ECs (HUVECs), acting as a protector against oxidative damage [36], while the expression of miR-200c was induced by H_2_O_2_ treatment in HUVECs, downregulating ZEB1 and increasing autophagy and senescence [37]. miR-494 is induced by endoplasmic reticulum (ER) stress in HUVECs, but this miRNA diminishes ER stress instead of increasing it. This is important since ER stress is another source of ROS that plays a key role in the progression of cardiovascular diseases [38,39] (Figure 2).

### 2.2. Effect of Oxidative Stress and ncRNAs in Macrophages during the Progression of Atherosclerosis

Endothelial dysfunction allows LDLs to enter the vascular wall and permits the recruitment of immune cells, such as monocytes, leading to inflammation and atherosclerotic progression [40].

The newly infiltrated monocytes differentiate into macrophages that capture ox-LDL, switching to foam cells, which precipitates mitochondrial and ER stress, increases ROS production, and leads to the progression of the disease [41]. Ox-LDLs have other effects in macrophages such as inducing the expressions of IL-12 and IL-18 and Th1 differentiation, as well as increasing the secretion of interferon gamma (IFN-γ) and, with this, the progression of atherosclerosis [42]. Additionally, as a more direct effect in macrophages, ox-LDLs promote the activation of NOXs and the subsequent increase in superoxide anions in these cells. All of this leads to an oxidation point in which the newly formed foam cells in the presence of even more LDL also have the capacity to oxidize them and further the malady [43] (Figure 2).

Mitochondrial OxS is also a source of ROS in macrophages; as a result, this stress activates the NF-кB signaling pathway, leading to an overexpression of monocyte chemotactic protein-1 (MCP-1) and recruiting more monocytes [44]. Furthermore, ROS production is not only important inside the macrophage, but also as signaling molecules outside the cell recruiting more inflammatory cells and affecting other vascular cells [45].

The molecular effect lncRNAs may have in macrophages during atherosclerosis has been a subject of study in the last years. LncRNAs mostly play a role in modulating ox-LDL-driven OxS in macrophages by sponging different miRNAs. Some examples are KCNQ1OT1, which sponges miR-137/tumor necrosis factor-α-induced protein 1 axe, increasing inflammation in THP-1 cells; GAS5, which sponges miR-135a and promotes the expressions of MDA, IL-1β, and IL-6 while decreasing SOD in THP-1 cells; and HOTAIR, which sponges miR-330-5p, augmenting inflammatory cytokines such as IL-6, IL-1β, and cyclo-oxygenase 2; in summary, when these lncRNAs are increased by the capture of ox-LDLs, they promote inflammation and OxS [46,47,48] (Figure 2). Other lncRNAs promote OxS by downregulating the expressions of antioxidant enzymes such as SOD. This was the case of NEAT1, an ncRNA that inhibited miR-128 and, therefore, increased IL-6, IL-1β, TNFα, and ROS [49]. The lncRNA UCA1 increased the pro-oxidant MDA, in this case by sponging miR-206, promoting ROS production and the expression of the CD36 receptor [50]. Another process that is also important is phenotype switching from an anti-inflammatory M2 to a pro-inflammatory M1 phenotype, and in this case, lncRNA260 was proved to be involved in promoting an M1 phenotype by downregulating the PI3K/AKT pathway and IL-28RA [51] (Figure 2).

There are other types of ncRNAs that also modulate OxS in macrophages in the presence of ox-LDL. The circRNA circTM7SF3 increased the expression of aspartyl/asparaginyl β-hydroxylase (ASPH) by downregulating the expression of miR-206 [52]. Some ncRNAs can be protective against the OxS and inflammation driven by ox-LDLs; for example, miRNA-135a was able to modulate foam cell formation, ROS production, and inflammation by increasing the expression of toll-like receptor 4 (TLR4) [53] (Figure 2). Moreover, miRNAs also affect macrophage polarization under different conditions; miR-21 was essential for macrophages to adopt an M1 phenotype in cardiac tissues [54] (Figure 2), and bone marrow macrophages secreted exosomes enriched in anti-inflammatory microRNAs, such as miR-99a, miR-146b, and miR-378a, which increased M2 macrophage polarization in recipient macrophages, diminishing the atherosclerotic injury of ApoE-deficient mice fed a western-type diet [55].

### 2.3. Effect of Oxidative Stress and ncRNAs in VSMCs during the Progression of Atherosclerosis

Vascular smooth muscle cells (VSMCs) are part of the media layer from the arteries, but during atherosclerosis and in the presence of ox-LDLs, they proliferate in an abnormal way and migrate to the intima layer where they deposit, forming a fibrous cap in the atherosclerotic plaque [56].

In response to different inflammatory and oxidative-driven chemokines, VSMCs can adopt different phenotypes, such as synthetic, osteochondrogenic, adipocytic, senescence, or foam cell types. More precisely, ROS seem to play a role in the switching to an osteochondrogenic phenotype, whereas ox-LDL is involved in switching to a foam cell phenotype (Figure 2).

During the progression of atherosclerotic injury, we can identify over eight types of plaques, but VSMCs are pivotal in the development of type-V, or fibroatheroma, plaque since is characterized by VSMC aberrant proliferation and migration to the subendothelial layer to form a cap above the lipid deposition. However, VSMCs are also important for the development of the following three types of injuries: type VI, or complicated fibroatheroma; type VII, or calcified lesions; and type VIII, or fibrotic lesions [57]. This calcification, defined by the deposition of hydroxyapatite mineral, is driven by cell apoptosis and oxidative stress and is important because it is linked to higher possibilities of plaque rupture [57,58].

As in other cell types, ncRNAs are also important in the role that VSMCs play in atherosclerotic progression. The lncRNA NCK1-AS1 was upregulated in plasma samples of patients with atherosclerosis and in ox-LDL-treated VSMCs, leading to an increase in inflammation and OxS; this effect could be due to the sponging of miR-1197 [59]. miR-4463 emerged as a modulator in VSMC phenotype switching under OxS driven by ox-LDL administration, and this effect was due to the regulation of the JNK and ERK pathways [60]. The migration and proliferation of VSMCs are key for plaque progression, and ncRNAs can be key factors for the modulation of these processes. For example, miR-31-5p is known to activate NOX enzymes and stimulate ROS production, as well as for proliferation and migration in VSMCs [61]. In fact, angiotensin-II, a molecule that may activate pro-inflammatory and pro-oxidant mechanisms, has been described to upregulate miR-31-5p [61,62]. miR-155 is also an activator of migration and proliferation in these cells, leading to neointimal hyperplasia. In this case, miR-155 mediated the targeting of mammalian sterile 20-like kinase 2 and increased inflammation and OxS, leading to vascular remodeling [63] (Figure 2).

### 2.4. Antioxidants and ncRNAs as a Treatment for Oxidative-Stress-Driven Vascular Injuries

Since there are a lot of mechanisms that drive vascular injuries during atherosclerotic progression and this disease is asymptomatic for a long period of time, there is a need to find new biomarkers for early detection of the disease, as well as possible treatments that diminish those processes that aggravate the disease, e.g., OxS.

Most of the approaches to reduce OxS during vascular damage have focused on antioxidant compounds able to diminish ROS production in the vascular wall; some examples are vitamins C, A, and E or polyphenols, such as resveratrol [5]. However, since ncRNAs have been proved to modulate different pathways of OxS in different cell types during atherosclerosis, they are emerging as potential targets for therapy.

The administration of miR-24 in balloon-injured diabetic rats reduced OxS by increasing the Nrf2/HO-1 pathway; this decreased ROS production while augmenting antioxidant enzymes such as SOD and glutathione peroxidase, resulting in the promotion of vascular endothelium repair [64]. miR-92a was upregulated in the aorta of db/db mice, and its downregulation by the administration of Ad-anti-miR-92a produced a decrease in ROS levels and an increase in HO-1 expression, with the inhibition of miR-92a having an atheroprotector effect [65]. Additionally, miR-140-5p was inhibited by the administration of a specific AntagomiR in mice treated with doxorubicin; this drug is famous for its cardiotoxicity by increasing OxS. However, a decrease in miR-140-5p expression increased Nrf2 levels and diminished mitochondrial stress and the injury generated by doxorubicin [66]. Finally, the inhibition of miR-351 in a diabetic mouse model with associated atherosclerosis was protective since this model had less lipid accumulation and cell apoptosis [67].

Other ncRNAs, such as CARMN, have been raised as promoters of atherosclerosis, participating in VSMC proliferation, migration, and lipid accumulation. In addition, a mouse model with depletion of CARMN showed wider injuries and a higher expression of pro-inflammatory proteins [68]. Antioxidant compounds may also have therapeutic effects by modulating ncRNAs. This was the case of notoginsenoside R1 (NR1), a naturally occurring compound that was used to treat ApoE deficient mice. Treatment with this compound decreased the atherosclerotic injuries in mice, as well as the OxS measured as circulating levels of SOD. Interestingly, NR1 treatment also affected the levels of different miRNAs: miR-21, miR-26a, and miR-126 were downregulated while miR-20a was increased, proposing these miRNAs as potential promoters or inhibitors during atherosclerosis progression. [69]. Moreover, ncRNAs are emerging as possible modulators of oxidative stress and atherosclerosis not only in preclinical studies. Currently, there is an ongoing clinical trial (Phase 2), to test the drug CDR132L, a selective inhibitor of miR-132, in heart failure and acute myocardial infarction.

Finally, ncRNAs are also found in plasma, e.g., miRNAs are found in plasmatic exosomes, and that is the reason they can also be used as biomarkers of vascular damage. In fact, very recently, a new study proposed miR-144 and miR-221 as biomarkers of atherosclerosis since both were overexpressed in the plasma of patients with atherosclerosis [70]. All these potential therapies are summarized in Table 1, indicating the expression changes described in the disease, the therapeutical approach selected to normalize the ncRNA levels, and the main outcome.

## 3. Oxidative Stress in NAFLD: Sources and Participation in Disease Progression

NAFLD is a hepatic disorder characterized by the ectopic accumulation of fat in the liver. According to recent data, it is estimated that the world prevalence of this disease is around 25% and that this prevalence may steadily increase in the near future [71]. Its progression ranges from benign hepatic steatosis to NASH, which can lead to the development of fibrosis, cirrhosis, and eventually, hepatocellular carcinoma (HCC) [72]. According to the widely accepted multiple hit theory to explain the onset and aggravation of NAFLD, hepatic OxS is crucial for progression to the later stages of the disease [73]. The mechanisms by which OxS are potentiated during NAFLD are diverse and include, among others, mitochondrial dysfunction, increased pro-oxidant enzyme expression, and impaired antioxidant protection.

The development of NAFLD involves an oversupply of FFAs that the liver cannot process, either by oxidation or by their conversion into triglycerides [73]. The main pathway for FFA oxidation is mitochondrial β-oxidation, which is overloaded in NAFLD and, therefore, leads to excessive electron flow through the ETC. Thus, the availability of electrons in the ETC is increased, and these are able to escape and react with oxygen in complexes other than cytochrome c oxidase [14,74]. In consequence, instead of water molecules, ROS are synthesized. ROS can later react with many types of biomolecules, including proteins, lipids, and DNA. As a mechanism to reduce ROS generation, hepatocytes increase the expression of UCP-2, which temporarily alleviates mitochondrial ROS generation; in turn, this situation depletes ATP in hepatocytes and energetically stresses them [75]. This process is known as mitochondrial dysfunction, in which mitochondria become much bigger in size and develop crystalline structures [76,77]. Mitochondrial dysfunction is related to increased mitochondrial membrane permeability, facilitating mitochondrial protein and ROS escape to the cytosol; therefore, this state promotes the propagation of OxS and induces cell death, aggravating liver injury in NAFLD [75].

In order to alleviate the deleterious effects of FFA accumulation, hepatocytes activate alternative pathways for lipid oxidation, such as microsomal ω-oxidation and peroxisomal β-oxidation. Specifically, in peroxisomes, these lipid species are processed by acyl-CoA oxidases (ACOXs), which generate hydrogen peroxide directly from catalytic activity. Therefore, peroxisomes themselves are a source of ROS in NAFLD. Moreover, the end-products of peroxisome activity are later directed to mitochondrial β-oxidation, thus contributing to the aggravation of mitochondrial dysfunction and mitochondrial β-oxidation overload [15]. In addition to ACOX activity, peroxisomes are organelles with a high content of iron, a fundamental cofactor for the processes that occur within this organelle. In the context of NAFLD, iron can leak from overloaded peroxisomes and directly catalyze ROS formation [78].

Other sources of ROS in NAFLD are NOX enzymes, which display a cell-specific expression distribution and are especially relevant to the function and activation of KCs and infiltrated macrophages. Immune cell activation during NAFLD progression induces the activation of NOX enzymes, which are directly involved in the production of superoxide anions [79]. More specifically, KCs are exposed to injury-related molecules (such as mitochondrial DNA) and lipopolysaccharides (LPSs) during NAFLD progression. LPSs induce the activation of TLR4, which eventually increases NOX2 activity and promotes ROS formation, thus activating NF-κB signaling. Moreover, the molecules released by damaged hepatocytes also converge in NF-κB activation, in turn aggravating the inflammatory response and eliciting ROS overproduction [80]. Although some studies have described increased expressions of NOXs in samples from animal models and humans with NAFLD, the involvement of these enzymes in OxS generation during NAFLD progression remains mostly unexplored. In line with the previous group of enzymes, inducible NOS (iNOS) is another marker of pro-inflammatory immune cell infiltration that directly contributes to the generation of ROS [81]. Finally, the amount of pro-oxidant molecules produced during NAFLD progression also affects HSCs, and in fact, ROS are well-recognized activators of HSCs [82]. Furthermore, HSCs produce ROS themselves when stimulated with profibrogenic factors, mainly TGF-β. These ROS further promote HSC activation and the progression towards fibrosis through the induction of collagen and α-smooth muscle actin [80].

As counterbalancing tools to regulate OxS, the liver possesses different antioxidant mechanisms. The Nrf2/Kelch-like ECH-associated protein 1 (KEAP1) system is a master regulator of antioxidant enzyme expression. Briefly, the association of KEAP1 with Nrf2 inhibits the translocation of Nrf2 to the nucleus in normal conditions; however, upon pro-oxidant signaling, this complex dissociates, and Nrf2 activates the transcription of antioxidant enzymes such as heme oxygenase 1 (HO-1) and NA(D)PH quinone dehydrogenase 1 (NQO1) [83]. Many studies have described that the transcriptional activity of Nrf2 is decreased in NAFLD, thus leading to decreased antioxidant protection [84]. Furthermore, the liver also contains antioxidant mechanisms, including GPX, glutathione S-transferases (GSTs), SOD, and catalase. Although many studies have indicated that the expressions of these enzymes are decreased in NAFLD, others have described the opposite [85,86,87,88,89]; therefore, it is likely that the antioxidant protection varies according to the stage of NAFLD.

3.1. ncRNA-Mediated Regulation of Oxidative Stress in NAFLD

It is well-known that ncRNAs regulate OxS in many cell types and diseases, including NAFLD. In fact, ncRNA-mediated regulation of OxS in the context of NAFLD has been described in most hepatic cell lineages (Figure 3).

Hepatocytes are the most abundant cell type in the liver, as well as the one mainly affected in NAFLD, and dysregulation of the miRNA expression in hepatocytes has been shown to promote OxS. For instance, miR-27a and miR-142-5p, which are verified regulators of antioxidant transcription factor Nrf2, were upregulated in an in vitro model of steatosis, and their inhibition led to an increase in Nrf2 expression, as well as increased antioxidant capacity [90]. Another group reported that miR-200a was downregulated in the livers of a rat model of NAFLD, impairing the regulatory effect on KEAP1 and leading to decreased expression of antioxidant mechanisms [91].

Peroxisomal β-oxidation can also be affected by aberrant miRNA expression. The upregulation of miR-540 in a genetic model of liver steatosis directly led to a decrease in the expression of peroxisomal proteins ACOX1 and PMP70, thus impairing peroxisomal function and aggravating NAFLD [92]. In the same way, miR-222 also downregulated ACOX1 and aggravated the progression of NAFLD [93]. In a large-scale study with human subjects, the serum expression of miR-193a-5p correlated with the hepatic expression of the antioxidant enzyme GPX8 in NAFLD patients, whereas lipid overload induced the downregulation of this miRNA, thus suggesting a regulatory mechanism involving this miRNA and antioxidant defenses in NAFLD [94]. Other reports suggested that increasing the expression of miR-29a could alleviate OxS during NAFLD progression, since transgenic mice overexpressing miR-29a and fed a steatogenic diet displayed lower DNA oxidation markers and antioxidant enzyme HO-1 expression. The amelioration of steatosis and OxS was attributed to the regulation of DNMT3b expression by miR-29a [95]. In agreement with this study, it was described that miR-29a downregulated the fatty acid importer CD36 and that miR-29a was downregulated in HFD-fed mice. CD36 promoted the internalization of FFAs, favoring mitochondrial dysfunction through the saturation of mitochondrial β-oxidation [96].

Other cell types also undergo OxS regulation mediated by miRNAs. In fact, miR-21 transcription is induced by NOX2 activation, which has a pro-oxidant role, and its expression induces M1 polarization in KCs [81,97]. However, miR-21 knockout mice fed with a fast-food diet displayed lower OxS markers than wild-type mice, suggesting that miR-21 itself might regulate OxS [98].

Regarding lncRNAs, LINC00240 was shown to be upregulated in NAFLD patients and FFA-stimulated hepatocytes, which, upon overexpression, increased ROS formation without affecting lipid accumulation [99]. Moreover, GAS5-mediated miR-26a downregulation was responsible for the induction of mitochondrial dysfunction and higher ROS production through blocking the regulatory effect of miR-26a on phosphodiesterase (PDE) 4B [100]. Remarkably, miR-26a was previously shown to protect from OxS triggered by FFA stimulation in cultured hepatocytes, likely due to the downregulation of PKCδ and the subsequent abrogation of NOX activation [101]. Other studies have also reported altered circRNA expression in hepatocytes and its relationship with OxS. In this sense, circRNA_00463667 reverted the loss of mitochondrial membrane potential induced by FFAs, thereby reducing the content of lipid peroxidation products and increasing SOD activity [102]. On the other hand, decreased circ_0048179 expression upon lipid overload exacerbated the downregulation of the antioxidant defense GPX4 mediated by miR-188-3p, thus aggravating OxS [103].

3.2. ncRNA-Based Therapies Targeting Oxidative Stress in NAFLD

Due to their ability to regulate numerous pathways, ncRNAs are becoming a very interesting tool to develop novel treatments targeting specific tissues (Figure 3). Specifically, for NAFLD, therapeutic strategies targeting the liver are able to selectively affect certain cell types and even subcellular organelles.

Preclinical studies commonly choose direct injection of an ncRNA inhibitor or mimic into the bloodstream in the absence of specific tissue targeting as a therapeutic approach to evaluate the effects in NAFLD animal models. For instance, inhibition of miR-665-3p with AntagomiRs in mice reverted the OxS caused by HFD feeding by inhibiting the downregulation of FDNC5, the precursor of irisin, which induced the activation of AMPKα [104]. AMPKα activation also reduced OxS in HFD-fed mice treated with miR-137-5p mimics, a miRNA that directly downregulated PDE4D expression and enhanced AMPK signaling [105]. Another miRNA that has been proposed as a therapeutic target for NAFLD is miR-103-3p, which targets ACOX1. By administering miR-103-3p antagonists to mice fed with a HFD, hepatic OxS was significantly ameliorated, along with liver histology and hepatic lipid content [106]. Adeno-associated viruses (AAVs) are a promising strategy for the management of liver diseases since the expression of a desired transgene can be maintained in the long term and does not require genomic integration [107,108,109]. Moreover, the existence of several serotypes allows for the specific transduction of genetic material in specific tissues, potentially enabling cell-specific expression of a transgene [107,108,109]. In the context of OxS quenching during NAFLD progression, Yang et al. developed serotype 8 AAV carrying a miR-802 inhibitor that simultaneously ameliorated insulin resistance, increased the expressions of antioxidant enzymes, and reduced ROS production [110].

Despite being much less abundant, RNA-based therapeutic strategies targeting lncRNAs can also modulate OxS during NAFLD progression. Indeed, inhibition of the lncRNA HULC in HFD-fed rats ameliorated the histological features of NAFLD while reducing lipid peroxidation and enhancing the antioxidant SOD activity [111]. The lncRNA RMRP directly regulated miR-206 and inhibited its activity on PTPN1, which aggravated NAFLD; therefore, Yin et al. demonstrated that RMRP inhibition in rats with NAFLD delayed the progression of the disease, significantly reduced MDA accumulation, and boosted the antioxidant defenses, as shown by increased SOD activity [112].

Although most strategies are oriented towards the modulation of OxS in hepatocytes, novel studies are also focusing on correcting this imbalance in the other resident cell types in the liver. Liver-macrophage-specific inhibition of miR-144, which is a validated regulator of antioxidant transcription factor Nrf2 expression, restored Nrf2 transcriptional activity and impaired ROS production. Surprisingly, this effect was also noticeable in hepatocytes, suggesting some type of cell-to-cell communication triggered by OxS [113]. In the later stages of NAFLD during its progression to HCC, the silencing of lncRNA SNHG20 in liver macrophages promoted their polarization to the M1 phenotype and promoted the expression of iNOS, a pro-oxidant enzyme that, in turn, increased OxS in the tumor and prevented the progression of tumorigenesis [114]. A very recent study elegantly overexpressed SCAR, a circRNA predominantly expressed in the mitochondrion of fibroblasts, in the liver using labeled nanoparticles targeting mitochondria. This strategy significantly reduced mitochondrial ROS formation and their escape to the cytoplasm by targeting ATP Synthase F1 Subunit Β (ATP5B), ultimately delaying the aggravation of NASH toward fibrosis, as well as improving the histological and metabolic manifestations of NAFLD [115]. A summary of all these potential therapies targeting OxS in NAFLD is shown in Table 2, indicating the expression changes described in the disease, the therapeutical approach selected to normalize the ncRNA levels, and the main outcome.

## 4. Future Perspectives on RNA-Based Therapies Modulating Oxidative Stress

As the global prevalence of obesity and its comorbidities rapidly increases, the urge to discover effective therapies to revert their deleterious effects is becoming critical [116,117]. The generation of OxS is crucial for the progression of all these diseases, and this state of oxidative injury is known to occur due to the appearance of a state of mild, sustained, whole-body inflammation [28]. Indeed, the direct targeting of OxS has gained substantial relevance over the last few years as a strategy to delay the progression of diseases such as atherosclerosis and NAFLD. Therefore, much effort is being made directed toward the management of antioxidant protection in the context of MS.

Traditionally, many of the studies in the field of OxS management focus on dietary interventions or direct supplementation with phytochemicals, mainly polyphenols, that directly suppress the generation of ROS and promote the increase in antioxidant mechanisms. Nonetheless, it has already been demonstrated that ncRNAs are also able to exert a post-transcriptional regulatory effect on the pro-oxidant and antioxidant proteins expressed in cells [21,22,28]. In fact, not only can ncRNAs be induced or repressed by the previously mentioned plant compounds, they can also constitute therapeutic targets themselves (Figure 4). This idea is reinforced by the fact that many RNA-based therapeutic strategies are being included in higher-phase clinical trials and that some of them have reached approval and commercialization by international drug agencies, as evidenced by drugs such as patisiran or RNA-based vaccines against SARS-CoV-2. However, no approved strategy exists using ncRNAs as therapeutic targets to date, as many of these clinical trials have yet to reach later phases in clinical trial progression [118].

Considering the current advances regarding the treatment of atherosclerosis using ncRNA-based therapies, miR-92a may be the candidate with the most evidence supporting its use as a therapeutic target. miR-92a has already been described as a promoter of eNOS uncoupling and subsequent OxS generation, and its inhibition with endothelium-targeting AAVs promotes the expression of antioxidant defenses [65,119,120]. Moreover, miR-92a was predicted to downregulate NOX enzymes, which maintained normal vascular function, providing some insight on the early events that promote OxS generation in atherosclerosis [121]. The role of miR-155, a pivotal miRNA in the pathophysiology of atherosclerosis, in OxS promotes it as a fundamental candidate for cell-specific modulation. For instance, in ECs, this miRNA promotes eNOS uncoupling and downregulates antioxidant protection, while its inhibition activates autophagy during OxS [36,122,123]. In macrophages, miR-155 inhibition promotes ROS generation by indirectly inducing NOX activation, whereas in VSMCs, OxS generation is driven by NOX upregulation [63,124]. Therefore, miR-155 vascular inhibitors can be expected to reach clinical trials in the near future.

Regarding NAFLD, many preclinical studies have suggested candidates for the liver-specific modulation of ncRNAs, although these studies do not specifically target these molecules in liver tissue. Only AAV-mediated miR-802 delivery is specifically targeted toward hepatocytes and appears to be the closest to clinical trials, despite not knowing the specific molecular mechanism that reduces OxS [110]. Another therapeutic strategy set to advance toward clinical trials is miR-144 inhibition in liver macrophages since preclinical studies suggested that it could delay NAFLD progression [113]. This study highlighted the importance of intercellular communication in NAFLD; in fact, other preclinical studies have highlighted the crucial role of intertissue communication through miRNAs in NAFLD, such as the BAT-transplantation-derived miR-96-5p inhibition of OxS by downregulating NOX4 [125]. Other promising strategies involve subcellular organelle targeting, such as mitochondrial-specific SCAR inhibition, which could easily be translated to clinical settings owing to efficient delivery through nanoparticles and the observed effects in the amelioration of NAFLD [115]. Nevertheless, it is fundamental to continue developing specific therapeutic strategies for the management of NAFLD that demonstrate safety and effectiveness.

Atherosclerosis and NAFLD are often concomitant since they are both related to obesity and MS, and in fact, individuals suffering from NAFLD have an increased risk of death due to cardiovascular events [126,127]. Therefore, finding pathogenic mechanisms shared by these pathologies is fundamental to developing multi-purposed therapies and simultaneously treating several diseases. Regarding the involvement of OxS in these diseases, some ncRNAs represent potential shared therapeutic targets. One of those targets is miR-206, which is downregulated in both diseases, although by different lncRNA regulatory mechanisms [50,112]. Thus, systemic treatment with miR-206 analogue RNAs could be beneficial to revert the pathogenic changes in vascular and hepatic tissues. Another ncRNA candidate for concerted regulation in both diseases is the overexpressed lncRNA GAS5, which downregulates miR-206 in NAFLD and miR-135a in atherosclerosis, leading to increased OxS [47,100]. Finally, miR-21 might represent a potential common target for both diseases since it has been related to the promotion of oxidative injury in models of both atherosclerosis and NAFLD, and its simultaneous inhibition in both tissues may represent an effective therapeutic strategy [54,98,128].

Despite the numerous regulatory axes that were mentioned throughout this review, there is still a long road ahead until ncRNA-based therapies reach clinical settings, at least in atherosclerosis and NAFLD. In fact, it was recently reported that liposome-encapsulated miR-34 mimic delivery for cancer treatment resulted in severe adverse effects and four deaths attributed directly to this treatment, most related to immune-driven toxicity. Despite these complications, most of the patients that were evaluable for response to the treatment presented stable disease or partial remission, which reinforced the idea that therapeutic strategies targeting ncRNAs could be effective for the clinical management of diseases [129]. Nevertheless, it is fundamental to continue researching in order to minimize the off-target effects by improving the delivery of RNA to tumors, as well as restricting immune system activation outside of the target organ. In addition, ncRNAs probably exert their regulatory effects on a multitude of targets at the same time. Therefore, it is likely that many of the ncRNAs that are altered during the progression of atherosclerosis and NAFLD could regulate the pro-oxidant–antioxidant balance in injured cells. This fact highlights the need to continue researching so that the mechanisms by which ncRNAs modulate OxS can be further elucidated. Not only can this allow the unraveling of more therapeutic targets, but it can also accelerate the approval and availability of effective therapies for these diseases, which remain without curative treatments.

## 5. Conclusions

In summary, OxS is fundamental to inducing the progression of atherosclerosis and NAFLD. As such, therapeutic targets that reduce OxS might be fundamental to ensuring better clinical management of these diseases. ncRNAs represent a promising tool to accurately regulate OxS, and the recent advances in RNA-based therapies suggest that ncRNA regulation might be of clinical utility in a few years. Recent examples of RNA-based therapies, such as vaccines conferring protection against SARS-CoV-2 infection, provide evidence that supports the feasibility and easy distribution of the use of RNA as drugs. In this sense, these therapies represent a starting point for the development of strategies involving ncRNAs, which still have to overcome challenges such as the specific targeting of genes. Nevertheless, there is still a great need to continue researching until the preclinical findings regarding ncRNAs constitute real clinical tools useful for the treatment of atherosclerosis and NAFLD.

## Figures and Tables

**Figure 1 antioxidants-12-00262-f001:**
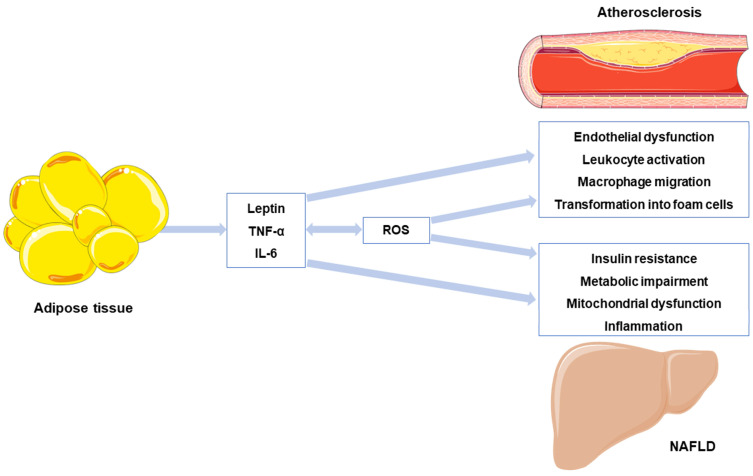
In the context of obesity and lipid overload, adipocytes acquire a proinflammatory phenotype that induces the secretion of adipokines, such as leptin, TNF-α, or IL-6. These adipokines promote ROS production and oxidative stress, which in turn stimulate proinflammatory adipokine secretion. Both types of cellular stress, inflammation and oxidative stress, play a role in the progression of atherosclerosis and non-alcoholic fatty liver disease (NAFLD), promoting several pathogenic processes. ROS: reactive oxygen species; TNF: tumor necrosis factor α; IL-6: interleukin 6. This figure has been edited from Servier Medical Art. Servier is licensed under a Creative Commons Attribution 3.0 Unported License.

**Figure 2 antioxidants-12-00262-f002:**
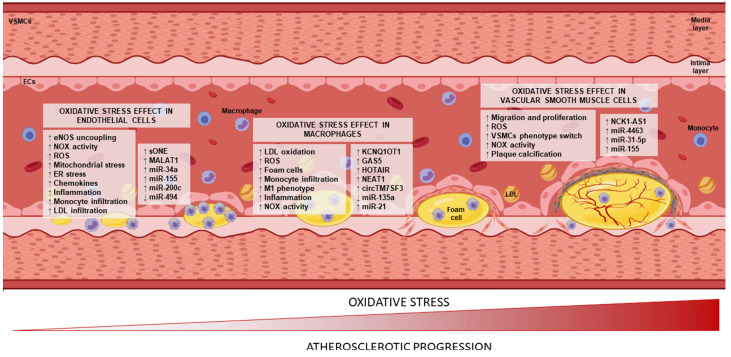
Roles ncRNAs and OxS play in atherosclerotic progression. The OxS effect in endothelial cells, macrophages, and vascular smooth muscle cells at a molecular scale and the regulation of different types of ncRNAs, such as sONE, GAS5, or miR-4463, respectively, are represented in this graphic. eNOS: endothelial nitric oxide synthase, NOXs: NADPH oxidases, ROS: reactive oxygen species, ER: endoplasmic reticulum, LDL: low-density lipoprotein, M1: type-1 macrophages, VSMCs: vascular smooth muscle cells. ↑ up regulation, ↓ down regulation.

**Figure 3 antioxidants-12-00262-f003:**
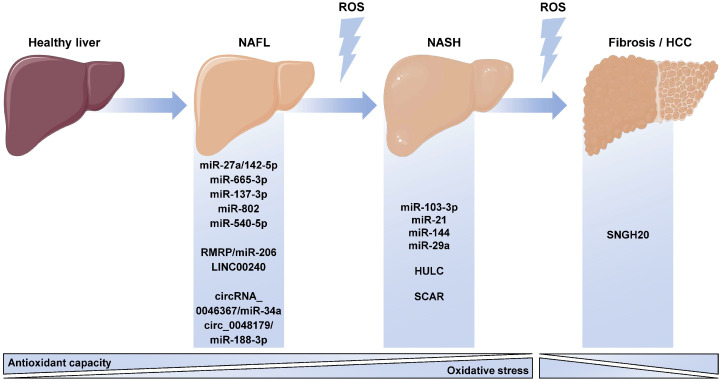
Each stage in the progression on NAFLD is related to disturbances in the pro-oxidant and antioxidant mechanisms, both of which are regulated by ncRNAs. In each stage, miRNAs, lncRNAs, and circRNAs that regulate oxidative stress and display altered expressions are highlighted. NAFL: non-alcoholic fatty liver, NASH: non-alcoholic steatohepatitis, ROS: reactive oxygen species, HCC: hepatocellular carcinoma. This figure has been edited from Servier Medical Art. Servier is licensed under a Creative Commons Attribution 3.0 Unported License.

**Figure 4 antioxidants-12-00262-f004:**
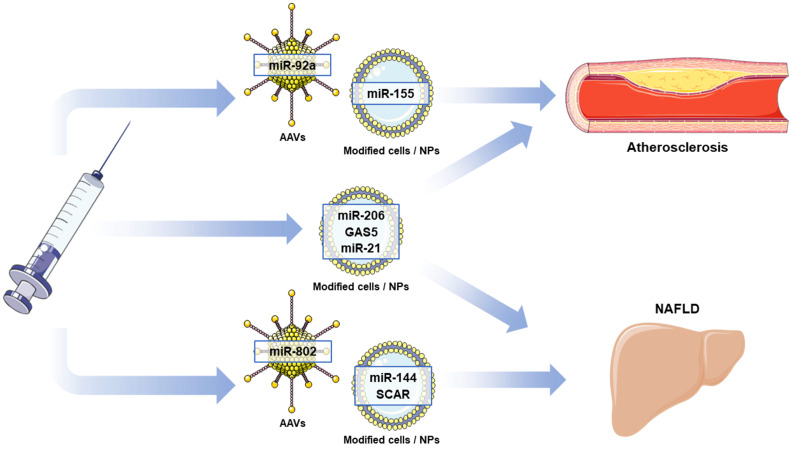
Potential ncRNAs that can be used as therapeutic targets for the modulation of oxidative stress in atherosclerosis and NAFLD. AAVs: adeno-associated viruses; NPs: nanoparticles; NAFLD: non-alcoholic fatty liver disease. This figure has been edited from Servier Medical Art. Servier is licensed under a Creative Commons Attribution 3.0 Unported License.

**Table 1 antioxidants-12-00262-t001:** Overview of ncRNA-based therapies for the treatment of oxidative-stress-driven injuries. ↑ up regulation, ↓ down regulation.

ncRNA	Expression Changes	Treatment	Molecular Mechanism	Biological Effect	References
miR-24	↓	Mimics	Activation of the Nrf2/HO-1 pathway	Reduction in OxS and promotion of vascular endothelium repair	[64]
miR-92a	↑	AntagomiR	Higher expression of HO-1	Reduction in OxS and atheroprotective effect	[65]
miR-140-5p	↑	AntagomiR	Increased Nrf2 levels	Alleviation of mitochondrial stress	[66]
miR-351	↑	shRNA	Higher expression of ITGB3	Reduction in lipid accumulation and apoptosis	[67]
CARMN	↑	Antisense oligonucleotides	Increased secretion of pro-inflammatory proteins	Wider atherosclerotic injury	[68]
miR-21	↑	NR1	Increased levels of circulating SOD	Reduction in atherosclerotic injury and OxS	[69]
miR-26a	↑				
miR-126	↑				
miR-20a	↓				
miR-144	↑			Potential use as biomarkers in plasmatic exosomes	[70]
miR-221	↑				

**Table 2 antioxidants-12-00262-t002:** Overview of ncRNA-based therapies for the assessment of NAFLD. ↑ up regulation, ↓ down regulation.

ncRNA	Expression Changes	Treatment	Molecular Mechanism	Biological Effect	References
miR-665-3p	↑	AntagomiR	Higher expression of FDNC5, activation of AMPK signaling	Reversion of OxS caused by HFD feeding	[104]
miR-137-5p	↓	Mimics	Downregulation of PDE4D expression and enhancement of AMPK signaling	Reduction in OxS	[105]
miR-103-3p	↑	AntagomiR	Increased ACOX1 levels	Reduction in OxS, improvement of NAFLD	[106]
miR-802	↑	miR-802 sponges	Lower activation of JNK and p38MAPK pathways	Reduction in OxS and improvement in insulin resistance	[110]
HULC	↑	siRNA	Increased SOD activity	Reduction in lipidic peroxidation and OxS	[111]
RMRP	↑	siRNA	Increased downregulation of PTPN1 by miR-206	Delay in the progression of NAFLD and boosted SOD activity	[112]
miR-144	↑	miRNA hairpin inhibitor	Restoration of normal transcription of Nrf2	Reduction in ROS production	[113]
SNHG20	↑	shRNA	Promotion of M1 polarization	Increment in OxS, prevention of tumorigenesis	[114]
SCAR	↑	shRNA and siRNA	Reduction in mitochondrial ROS formation by targeting ATP5B	Delay in the aggravation of NASH	[115]

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
