# Peer review of "Oxidative Stress Modulation by ncRNAs and Their Emerging Role as Therapeutic Targets in Atherosclerosis and Non-Alcoholic Fatty Liver Disease"

_antioxidants, 2023, doi:10.3390/antiox12020262_

Round 1

Reviewer 1 Report

antioxidants-2144759-peer-review-v1

The review work presented by Jorge Infante-Menéndez and co-workers titled “Oxidative Stress Modulation by ncRNAs and their Emerging Role as Therapeutic Targets in Atherosclerosis and Non-Alcoholic Fatty Liver Disease” is well written, clear, and easy to read. The topic is very interesting; therefore, it adds clustered information to the innovative therapeutic options for Oxidative Stress mediated disease linked to obesity. This aspect is a cutting-edge area, and we still do not have a definitive radical pharmacological intervention. The author performed a very well-conceived overview of the role of Oxidative Stress modulated non-coding RNA focused on only one aspect, as it is stated in the title.  

My suggestion is to rewrite the conclusion, implementing this section. Today a mRNA-based vaccine exists. 

Reviewer 2 Report

Authors reviewed the role of ncRNAs as therapeutic targets in atherosclerosis and non-alcoholic fatty liver disease. This is a well-reviewed paper and each section is well organized with figure.

Please add a table in 2.4 section to summerize ncRNAs, up/down regulated, antago or mimics, its role or mechanism, and reference, etc. (for example, about miR-92a, miR-140-5p, miR-351, CARMN, miR-21, miR-26a, miR-126, miR-20a miR-144 and miR-221)

Please add a table in 3.2 section to summerize ncRNAs, up/down regulated, antago or mimics, its role or mechanism, and reference, etc. (for example, about miR-665-3p, miR-137-5p, miR-103-3p, miR-802, HULC, RMRP/miR-206, miR-144, SNHG20, SCAR)

Recently, several ncRNA-based therapeutics have been developed and are currently entered into different phases of clinical trials. For the treatment of vascular disease, miR-143/145 and miR-92 are in Preclinical stage. For the treatment of nonalcoholic fatty liver disease, miR-103/107 is in clinical trial phase 2. Please discuss ncRNA-based therapeutics for the treatment of atherosclerosis in clinical trial or preclinical stage.

Effects of miRNA therapeutics are not necessarily restricted to the intended tissue or cells but can also cause systemic side effects. Recently, a prominent example of the occurrence of disastrous side effects is MRX34, a synthetic miR-34a mimic. So, please discuss the side effects of miRNA therapeutics.

Reviewer 3 Report

I think this review is comprehensive. A few minor points that could add to the review I suggest: 1. In the sections about NAFLD, the authors seem to focus on hepatocytes. As inflammation is highly involved in the NAFLD progression, how does oxidative stress affect non-parenchymal cells, such as Kupffer cells and stellate cells could be discussed. 2. The authors focus on two diseases in this review: NAFLD and atherosclerosis. Although the authors discussed in the introduction that both diseases are related to obesity and oxidative stress is implicated in obesity, the specific reasons why authors focus on these two diseases is not very clear, as there are many other diseases are related to obesity. I think it would be better to emphasize why oxidative stress is particularly implicated in these two diseases, and maybe to compare and contrast the effect of oxidative stress in these two diseases as well. 

Round 2

Reviewer 2 Report

Authors reviewed the role of ncRNAs as therapeutic targets in atherosclerosis and non-alcoholic fatty liver disease. This is a well-reviewed paper and each section is well organized with figure. And the manuscript was revised by providing a point-by-point response to the reviewer's suggestion.